# Surgical Margins in Canine Cutaneous Soft-Tissue Sarcomas: A Dichotomous Classification System Does Not Accurately Predict the Risk of Local Recurrence

**DOI:** 10.3390/ani11082367

**Published:** 2021-08-11

**Authors:** Lavinia Elena Chiti, Roberta Ferrari, Paola Roccabianca, Patrizia Boracchi, Francesco Godizzi, Giuseppe Achille Busca, Damiano Stefanello

**Affiliations:** 1Dipartimento di Medicina Veterinaria, Università degli Studi di Milano, 26900 Lodi, Italy; lavinia.chiti@unimi.it (L.E.C.); paola.roccabianca@unimi.it (P.R.); francesco.godizzi@unimi.it (F.G.); damiano.stefanello@unimi.it (D.S.); 2Laboratorio di Statistica Medica, Biometrica ed Epidemiologia “A. Maccaro”, Dipartimento di Scienze Cliniche e di Comunità, Università degli Studi di Milano, 20122 Milan, Italy; patrizia.boracchi@unimi.it; 3Veterinary Teaching Hospital, Università degli Studi di Milano, 26900 Lodi, Italy; giuseppe.busca@unimi.it

**Keywords:** dog, sarcoma, margins, surgery, recurrence

## Abstract

**Simple Summary:**

Histological evaluation of surgical margins is crucial for correct prognostication and adjuvant treatment recommendation after excision of soft tissue sarcoma (STS) in dogs. Incompletely excised STS have a high risk of local recurrence (LR), while completely excised STS without other negative prognostic factors are generally associated with a good prognosis. However, guidelines are lacking on how to manage STS excised with clean but close margins (CbCM), although some authors advocated their inclusion in the tumor-free margin group. This retrospective study investigates the impact of CbCM on LR of canine STS. Ninety-eight surgical excised canine STS at first presentation were included. Cumulative incidence of LR was estimated for each category of margins (tumor-free, infiltrated, CbCM), and after grouping CbCM alternatively in the tumor-free and infiltrated category. Cumulative incidence of LR at three years differed significantly between the three categories, and it was estimated to be 42% with infiltrated margins, 23% with CbCM, 7% with tumor-free margins. Both when CbCM were grouped with infiltrated margins or with tumor-free margins, the incidence of LR was statistically different. The rate of LR with CbCm was greater than with tumor-free margins. The category CbCM may be considered as a separate prognostic category.

**Abstract:**

Adjuvant treatments are recommended in dogs with incompletely excised cutaneous soft-tissue sarcoma (STS) to reduce the risk of local recurrence (LR), although guidelines are lacking on how to manage clean but close margins (CbCM). This retrospective study investigates the impact of CbCM on LR of canine STS. Ninety-eight surgically excised canine STS at first presentation were included. Tissue samples were routinely trimmed and analyzed. Cumulative incidence of LR was estimated for each category of margins (tumor-free, infiltrated, CbCM), and included CbCM in the tumor-free and infiltrated category, respectively. The prognostic impact on LR was then adjusted for relevant prognostic factors. Cumulative incidence of LR at three years differed significantly between the three categories (*p* = 0.016), and was estimated to be 42% with infiltrated margins, 23% with CbCM, 7% with tumor-free margins. Both when CbCM were grouped with infiltrated margins (*p* = 0.033; HR = 5.05), and when CbCM were grouped with tumor-free margins (*p* = 0.011; HR = 3.13), a significant difference between groups was found. STS excised with infiltrated margins had the greatest risk of LR. The rate of LR with CbCm was greater than recurrence rate of tumor-free margins. The category CbCM may be considered as a separate prognostic category.

## 1. Introduction

Canine soft-tissue sarcomas (STS) display a range of biological behaviors with tumor grade being regarded as one of the main predictors of local recurrence (LR) and of development of distant metastases [1,2,3]. Given their propensity for local invasiveness but their relatively low rate of distant spread, complete surgical excision is crucial to achieve long-term disease control [1,4,5,6,7,8]. A recent metanalysis on canine STS confirmed the impact of surgical margins on LR, with LR rates of 9.8% for STS with tumor-free margins and 33.3% with infiltrated margins, with a significantly lower risk of LR for STS excised with tumor-free margins (overall relative risk of 0.4) [7]. Consensus on margin assessment and reporting are still lacking for canine STS, with different studies considering different and arbitrary cut-offs for histologic safe margins and mostly without validated correlation with prognosis [5,7,9,10,11,12,13,14,15,16].

The consensus statement of veterinary surgical pathologists by Kamstock and colleagues (2010) suggests the application of objective measurements of the distance between the tumor limits and the surgical tissue edges to objectively assess histological margins [17]. However, clinical studies on canine STS report the use of qualitative criteria for margin categorization, with dichotomous and trichotomous classifications being the most commonly used [13,16,17]. In the dichotomous system, histological margins are classified as tumor-free or infiltrated, while the trichotomous system embodies an additional intermediate category defined as “narrow” or “clean but close” margins (CbCM) [1,4,5,10,11,18,19,20]. Discrepancies in margin assessment of surgically excised STS have led to conflicting results, with rates of LR ranging from 0% to 14% for cases with tumor-free margins and from 17% to 75% with infiltrated margins [1,2,3,5,10,19,20]. Additionally, cases included in the CbCM category are reported as being inconsistently at risk or not for LR [7,10,11]. Hence, identification of those dogs that may benefit from local adjuvant treatments to prevent LR following STS surgical excision is not straightforward in the instance of CbCM, with a high risk of under- or overtreatment [15,16].

The use of a simplified dichotomous classification with a cut-off for histological tumor-free margins (HTMF) >0 mm has been recently advocated for categorization of histological margins in canine STS [15]. This classification is reportedly predictive of LR for STS in humans treated with surgery and adjuvant radiation therapy (RT), implicating a lack of prognostic significance of CbCM [3,15,21,22]. However, the actual prognostic impact on LR of CbCM in surgically excised canine STS has not been elucidated yet in a clinical setting. Hence, this retrospective study aims at investigating the impact of CbCM on LR of surgically excised STS in a large mono-institutional cohort of dogs. We hypothesized that CbCM would not display a significantly different behavior from tumor-free margins, and that a simplified and more objective dichotomous classification with a cut-off of HTFM > 0 could better correlate with prognosis and thus be consistently applied for a simplified categorization of histological margins of canine STS.

## 2. Materials and Methods

This was a mono-institutional retrospective (2001–2019) study. Inclusion criteria for STS cases were as follows: dogs with histologically confirmed cutaneous and subcutaneous STS at initial presentation, without loco-regional and/or distant metastases (excluded with preoperative fine needle aspiration of any enlarged lymph node, abdominal ultrasound and thoracic radiographs or whole-body contrast-enhanced CT scans), treated with curative-intent surgical excision with or without adjuvant chemotherapy. Exclusion criteria were a diagnosis of visceral STS, recurrent STS, ongoing neoadjuvant chemotherapy or adjuvant and/or neoadjuvant RT at presentation/diagnosis.

Data including signalment, tumor aspects (location, size at the longest diameter, ulceration), type of surgery, histopathological report and adjuvant chemotherapy (when administered) were retrieved from clinical records. Tumor sites were grouped as follows: distal limbs (below the elbow or stifle joint), proximal limb, head and neck, thoracic wall, abdominal wall; for statistical purposes tumors were defined as located on the distal limbs or elsewhere as previously reported [10].

Curative-intent surgical excision was planned as the widest possible in relation to tumor site and characteristics, and classified based on the Enneking system into marginal (surgical margin adjacent to pseudocapsule), wide local (2–3 cm of lateral margins of healthy tissue and 1–2 deep fascial planes) or radical (excision of the entire compartment) [9,23].

Immediately after excision, the specimens were spatially orientated by the surgeon with suture tags and submitted for histopathology. One board-certified pathologist (P.R.) and a resident in training (F.G.) examined all the excised specimens. Histopathological reports included microscopic description, grade (9), mitotic count (area of 2.37 mm^2^), percentage of necrosis (absent, <50%, >50%) and pattern of tumor growth (expansile or infiltrative). Surgical margins were evaluated by radial sectioning [17,24]. Pathologists measured the mm between the tumor and the surgical excisional cut at the narrowest point on paraffin-fixed specimen and classified margins based on a trichotomous system in:-Tumor-free: HTMF > 3 mm-CbCM: HTMF 1–3 mm-Infiltrated: neoplastic cells on the surgical cut (“tumor on ink”) [15]

Follow-ups were collected during periodical clinical rechecks for the first two years, scheduled at different time lapses for each dog based on the indications of the attending oncologist, and by telephone conversations with the owner or referring veterinarian thereafter. Disease progression (if any) and status of the dog (alive or dead) were recorded during follow-ups. Local recurrence was defined as cytologically or histologically confirmed STS regrowth within 2 cm from the previous surgical scar; loco-regional relapse was defined as regional nodal metastases confirmed by cytology or histology and distant relapse was defined as metastases to any distant site confirmed by diagnostic imaging and/or cytological/histopathological findings. Cause of death was further defined as tumor-related if spontaneous death occurred or humane euthanasia was elected due to tumor progression, and as tumor-unrelated otherwise. Time to LR (TLR) was calculated from the day of surgery to the day of LR. Dogs lost to follow-up before LR were censored at the date of the last contact.

### Statistical Analysis

Local recurrence was set as the main end point for analysis. Since death could prevent the observation of LR, a method for competing risk was used to estimate the cumulative incidence of LR in the whole sample and in the three categories of histological margins (tumor-free, CbCM, infiltrated), and after grouping CbCM alternatively with tumor-free and infiltrated margins. The cumulative incidence curves were then compared with a Gray’s test. [25] The impact on TLR of surgical margins classified with both dichotomous classification systems (tumor-free margins + CbCM versus infiltrated margins, tumor-free versus infiltrated margins + CbCM) was then adjusted for other well-known prognostic variables including tumor location [20] and size [2,5,20], tumor grade [1,2,26], mitotic index [27,28], pattern of growth [5], histotype [5,20,28] and adjuvant chemotherapy by the Fine and Gray semiparametric regression model, an extension of Cox model suitable for competing risk [29]. Results are reported as sub-distribution hazard ratios with 95% confidence intervals and *p*-value of Wald test. Margins and other categorical prognostic factors were included in the model as dummy variables and continuous variables were included in their original measurement scale. Although a multivariable regression model would have been the preferred approach to adjust the prognostic effects of margins for the other prognostic variables, this was not possible due to the low number of events [30], thus the prognostic effect of margins was adjusted separately for each variable. Median follow-up time was calculated with a reverse Kaplan–Meier method.

Statistical analyses were performed using a software package (R—software; www.r-project.org [accessed on 10 December 2020], with library cmprsk for competing risk analysis). The significance level was set at 5%.

## 3. Results

Ninety-four dogs were included in the study, with a total of 98 STS; four dogs had two STS each. Breeds were distributed as follows: 35 (37.2%) mixed-breeds, 9 (9.6%) Boxers, 6 (6.4%) Labrador Retrievers, 6 (6.4%) German Shepherds, 5 (5.3%) Dobermann Pinschers, 4 (4.3%) Jack Russel Terriers, 4 (4.3%) Rottweilers, 3 (3.2%) Cavalier King Charles Spaniels, 3 (3.2%) Siberian Huskies, 2 (2.1%) each of Bretons, Golder Retrievers, Belgian Shepherds and one each of American Staffordshire, Dachshund, Beagle, Italian Hund, Maltese, Pinscher, Pitbull, Pointer, Schnauzer, English Setter, Pomeranian, and Whippet. There were 41 (43.6%) intact males, 34 (36.2%) spayed females, 11 (11.7%) intact females and 8 (8.5%) neutered males. Median age at presentation was 10 years (range 2.7–17 years), and median weight was 25 kg (range 3.5–52 kg); weight was not available in three dogs.

Forty-nine STS were located on the proximal limbs (50%), 24 (24.5%) on distal limb (below elbow and knee joint), 11 (11.2%) on the head and neck, 8 (8.2%) on the thoracic wall and 6 (6.1%) on the abdominal wall. Tumor clinical dimensions at the widest diameter were available in 94 cases (96%) and ranged from 0.5 to 15 cm (median: 5 cm). Nine (8.2%) STS were ulcerated and 88 (90.8%) were covered by intact skin; information on status of overlying skin were not available for one tumor. Surgical excision was marginal in 76 (77.5%) cases, wide in 18 (18.4%) cases and radical in 3 (3.1%) cases; in one case this information was not available.

At histopathology, 78 (79.6%) tumors were diagnosed as perivascular wall tumors (PWT), the other 20 (20.4%) cases were comprised of fibrosarcoma (n = 4), fibromixosarcoma (n = 3), undifferentiated sarcoma (n = 3), rabdomyosarcoma (n = 3), mixosarcoma (n = 2), nerve sheath tumor (n = 2) and liposarcoma (n = 3). Sixty (61.2%) STS were grade I, 33 (34.7%) were grade II and 5 (5.2%) were grade III. Median mitotic count was two (range 0–40). Tumor necrosis was absent in 60 (63.2%) tumors, <50% in 28 (29.5%), >50% in 7 (7.4%), and unavailable in three cases. Pattern of growth was available in 77 tumors (60 PWTs, 17 other histotypes), 38 (49.4%) of which were expansile and 39 (50.6%) infiltrative. Tumors were excised with tumor-free margins in 29 (29.6%) cases, CbCM in 24 (24.5%) cases and with infiltrated margins in 45 (45.9%). Of the 24 CbCM, 16 resulted from marginal and eight from wide/radical surgical excision.

Adjuvant chemotherapy was administered to 17 (18%) dogs, using one of the following protocols: metronomic chemotherapy with oral cyclophosphamide (n = 14), Doxorubicin alone (n = 2) and Doxorubicin followed by metronomic cyclophosphamide (n = 1).

At the end of the study, 12 dogs were alive (follow-up time: 122–1695 days); of these, two had developed LR. Fifty-eight dogs had died, 14 of which for tumor related causes: seven with LR, two with LR and distant metastases and five with distant metastases. Of the 44 dogs that died of tumor unrelated causes, seven had had LR that was successfully treated. Twenty-four dogs were lost to follow-up, five of which after detecting LR (follow-up 30–595 days) and 11 before detecting LR (follow-up 15–365 days) (Table 1). Median follow-up time was 1425 days.

During the follow-up period, a total of 23 dogs developed LR. The cumulative incidence of LR was 12% at six months (95% Confidence Interval [C.I.] 5.30–18.69%), 19% at one year (95% C.I. 10.70–26.95%), 25% at two years (95% C.I. 16.05–34.68%) and 26.9% at three years (96% C.I. 17.28–36.48%) (Figure 1).

Cumulative incidence of LR at three years of follow-up was 41.2% in dogs with infiltrated margins, 22.5% with CbCM and in 7.1% with tumor-free margins. The difference among the cumulative incidence curves with the trichotomous classification was statistically significant (Gray test = 8.23, *p* = 0.016) (Figure 2). The hazard of developing LR was 1.92 times higher with infiltrated margins than CbCM (95% C.I. 0.71–5.17), 3.21 times higher with CbCM than tumor-free margins (95% C.I. 0.60–17.1) and 6.16 times higher with infiltrated than tumor-free margins (95% C.I. 1.37–27.8). The difference was statistically significant only between infiltrated and tumor-free margins (*p* = 0.018), while it was not significant for CbCM and tumor-free margins (*p* = 0.170), and for CbCM and infiltrated margins (*p* = 0.2).

To assess which dichotomic classification may be best suited to predict the risk of LR, cumulative incidence curves were compared after grouping the CbCM cases either with the tumor-free or the infiltrated margin categories. There was a statistically significant difference both between tumor-free margins + CbCM versus infiltrated margins (*p* = 0.011; HR = 3.13, 95% C.I. 1.29–7.59) and between tumor-free margins versus CbCM + infiltrated margins (*p* = 0.033; HR = 5.05, 95% C.I. 1.14–22.4) (Figure 3).

The first classification (tumor-free margins + CbCM versus infiltrated margins) remained significant when adjusted for histotype, tumor grade, mitotic count, pattern of growth, tumor site, tumor size and adjuvant treatment (Table 2). Conversely, when CbCM were grouped with infiltrated margins, the comparison with tumor-free margins remained significant only when adjusted for mitotic count, tumor type and site, while no significance was detected when the conjunct impact of tumor grade, pattern of growth, tumor size and adjuvant treatments was evaluated (Table 2).

## 4. Discussion

Although it is widely accepted that surgical excision with histologically tumor-free margins is protective against LR of canine STS, the relevance of margin width has been rarely investigated in veterinary oncology and has not been previously assessed specifically for canine STS [5,6,7,11,19,27,31]. To the best of the authors’ knowledge, only one study has explored the impact of CbCM on LR of cutaneous malignancies, including STS, in dogs and cats demonstrating a significant difference in the rate of LR between tumors excised with CbCM and tumor-free margins [31]. Likewise, in the present study, STS excised with CbCM had an LR rate at three years significantly higher than those excised with tumor-free margins, suggesting that the width of the histological margins may bear some impact on the risk of LR. Although the hazard of developing LR was statistically different only between tumor-free and infiltrated margins, a tendency towards significance was found also when comparing CbCM with tumor-free margins, with a three-fold increase in the risk of LR in dogs with CbCM versus tumor-free margins. The incidence of LR in the sample population was not similar between CbCM and infiltrated margins either, with STS excised with infiltrated margins having a hazard of LR two times higher than those with CbCM. Hence, according to our results, categorization of margins based on a simplified dichotomous system may not be optimal to accurately predict the risk of LR following curative-intent surgery for canine STS.

Given the lack of standardization of histological margin assessment, it is difficult to compare the impact of histological margin width on LR after surgical excision observed in this caseload with previous studies on canine STS. A recent systematic literature review highlighted the variability of margin classification among studies [16], pointing out the application of different classification systems and arbitrary cut-offs for each category; of 11 papers dealing with STS, five applied a dichotomous and six a trichotomous classification system, with cut-offs varying between 0 to 10 mm to indicate complete margins and/or 1 to 3 mm for CbCM [16]. The assessment of the actual risk of LR based on the existing literature may be particularly challenging for STS excised with CbCM, considering that even in studies that apply a trichotomous system for histological margin classification, CbCM are arbitrarily included either in the tumor-free or infiltrated margin category for LR risk analysis [7,10,11,31]. This approach may cause an under or over estimation of the risk of LR for CbCM cases and potentially leads to unfit treatment recommendations, especially when considering that there has been little evidence obtained by studies on the determination of the significance of CbCM and if these cases should be considered as those with tumor-free or infiltrated margins [15].

Given the variability in margin assessment and lack of strong evidence of a different behavior of close or wide margins, the adoption of a simplified dichotomous classification has been suggested to facilitate the standardization of histological margin assessment for canine STS by the application of the R classification [15]. The R classification is increasingly utilized in human oncology and defines R0 (complete excision) as a histologic tumor free margin (HTMF) >0 mm, R1 (incomplete excision) as “tumor on ink” and R2 as an intralesional excision [32]. This classification is based on the assumption that only tumors that reach the inked margin (R1) have an elevated risk of LR, whilst the distance from tumor to margin does not significantly impact the rate of LR in microscopically tumor-free margins (R0) [21]. The American Joint Committee on Cancer (AJCC) has promoted the use of R classification in human oncology to improve uniformity in margin reporting and to facilitate communication between clinicians, researchers and with patients. The main flow in adapting R classification to canine STS is that surgery is often the only treatment modality applied in dogs, whilst the human counterpart is most commonly treated with surgery and adjuvant RT, potentially improving the control of neoplastic cell clusters arising near the margin edge in CbCM [33].

Indeed, the importance of adjuvant RT for local tumor control is well established in human medicine, supported by the evidence of sarcoma cells being found up to 4 cm away from the primary tumor, and categorization of LR rate based on margin status with or without RT may significantly differ [34]. Arguably, while evidence has been produced that STS excised with wider margins have a similar LR rate that those with CbCM in humans when adjuvant RT is consistently applied, in one study where only 39% of patients received adjuvant RT, the LR rate after surgical excision of STS was significantly higher for <10 mm excisional margins compared to >10 mm margins [35]. Similarly, a recent study on non-infiltrative soft-tissue sarcomas suggested using 5 mm as the cut-off for safety margins in people treated with surgery alone [36].

In dogs, adjuvant RT does also improve local control of incompletely excised STS, with reportedly prolonged survival times and rates of LR lower than 20% [26,37,38]. However, RT is not widely available in veterinary oncology due to the high costs and need for dedicated personnel and facilities, and in this scenario the use of a simplified R classification system may underestimate the risk of LR for STS excised with narrow margins, thus potentially undertreating dogs.

To complicate things further, canine STS display a wide range of histological patterns and heterogenous clinical behaviors and may thus recur or not independently from completeness of excision, as suggested by the highly variable rate of LR reported for both histologically tumor-free and infiltrated margins [2,3,7,10]. Several prognostic variables have been described alongside margin status, including histopathological and clinical features [1,5,6,20,27,39]. Hence, in the present study, the impact of margin status was adjusted for relevant prognostic variables after grouping CbCM alternatively with tumor-free or infiltrated margins, although it should be emphasized that evaluation of the impact of variables other than margin status on LR was beyond our aim, and variable selection for multivariate models was thus dictated by previous evidence of their prognostic significance [1,5,7,20,27,39].

The fact that incidence of LR differed statistically between margin categories both when regarding CbCM as tumor-free or infiltrated (CbCM + tumor-free vs. infiltrated; tumor-free vs. infiltrated + CbCM) is not surprising, given that both studies that included CbCM in the tumor-free category and in the infiltrated category have been previously able to demonstrate an impact of surgical margins on LR [7,11,20]. However, in multivariate analysis on the study population, margin status was independently prognostic for LR only when CbCM were grouped with tumor free cases, while significance was partially lost if CbCM were joined with the infiltrated margin cases, suggesting that CbCM STS have a closer connection to tumor-free STS than those with infiltrated margins. It is reasonable to assume that the impact on LR of infiltrated margins was demoted by the reduced negative effect of CbCM when CbCM were merged with infiltrated margins, thus causing margin status to lose its impact on LR when adjusted for more significant prognostic factors, such as tumor grade. This result is in contrast with the study by Scarpa and colleagues, that reported a higher accuracy in predicting LR of various cutaneous malignancies when combining CbCM with infiltrated margins [31]. The main limitation of the latter study was the inclusion of multiple canine and feline tumor types with highly variable biological behavior, which precluded the adjustment of the impact of margins status for tumor-specific prognostic variables and may explain the discrepancy with our results [15,31].

In the present study, 17 dogs received adjuvant chemotherapy, with all but two of them receiving metronomic cyclophosphamide alone or in combination with doxorubicin. Although evidence of the efficacy of traditional chemotherapy is limited, metronomic chemotherapy seems to reduce the risk of LR of incompletely excised STS [40]. Hence, we adjusted the impact of histological margins for adjuvant treatment in order to evaluate the potential confounding effect of different treatment modalities. When CbCM were classified together with infiltrated margins, adjuvant treatments were prognostic for LR, but margin status lost its significance, whilst when CbCM were included with the tumor-free category, margin status and adjuvant therapies remained both prognostic for LR. This result underscores the assumption that, if adjuvant treatment is given, CbCM cases tend to behave similarly to tumor-free margins, despite the higher LR rate observed for CbCM when applying the trichotomous classification. Future studies should evaluate the efficacy of adjuvant chemotherapy, and especially of metronomic cyclophosphamide, in preventing LR of STS excised with CbCM in order to determine whether this treatment modality could be a valid alternative to RT.

Another potential source of confusion in assessment of tumor margins is the lack of standardized guidelines for trimming methods in the veterinary literature. Indeed, the use of different trimming methods may influence the accuracy of margin evaluation. In the consensus statement of veterinary pathologists, it has been suggested that trimming techniques that allow for examination of a greater percentage of marginal tissue may most likely result in a more accurate prediction of the actual status of excisional margins [17]. It has been previously hypothesized that differences in the LR rates between CbCM and tumor-free margins may be due to examination of an insufficient amount of tissue at the margin edge, thus leading to incorrect inclusion of some infiltrated margins in the CbCM category [41]. In the present study, however, a combination of cross-sectioning and tangential sectioning was applied for margin evaluation in order to examine the widest surface of margin edges feasible to yield a reliable margin assessment.

Notably, PWTs represented almost 80% of STS included in this caseload. Although PWTs have been recognized as a subgroup of STS with a less aggressive behavior, the actual prevalence of this histotype has not yet been investigated [5,20,42]. Thus, the relatively low rate of LR and long-term survival of this caseload may be due to the high prevalence of PWTs, although, surprisingly, histotype did not show significance for LR prognosis when evaluated jointly with histological margin status. However, to further confirm this result, future investigations including a higher number of STS of different histotypes than PWT should be performed.

This study has some limitations. First, the inclusion of STS with different histotypes and histological grades may have been a source of bias when evaluating the impact of margin status on LR, given the prognostic significance of those variables [1,2,5,20,26,28]. Although margin status was adjusted for these prognostic variables in bivariate models, it was not possible to perform a multivariate model including all the variables, due to the relatively low number of LR that did not allow for further sample stratification. Moreover, the inclusion of animals that received adjuvant chemotherapy may have influenced patients’ outcome, potentially leading to underestimation of the rate of LR. Future prospective studies should evaluate the accuracy of different cut-offs of HTFM in predicting LR of canine STS stratified by treatment modality and prognostic variables, in order to establish the width of the histological safety margins and improve margin reporting in veterinary oncology.

## 5. Conclusions

In conclusion, results of the present study suggest that wider histological margins provide better local control after surgical excision of STS in dogs, with 23% versus 7% LR rates reported for STS excised with CbCM (HTMF < 3 mm) and tumor-free margins (HTMF > 3 mm), respectively, and the hazard of LR increasing by three-fold with CbCM.

Contrary to the initial working hypothesis, the use of a simplified dichotomous classification may underestimate the risk of LR, potentially not allowing for correct identification of animals that may benefit from adjuvant treatments. The lack of LR in 77% of STS excised with CbCM and in 58% of infiltrated margins, however, underscores the assumption that recommendation of adjuvant treatment should take into account the interaction between multiple prognostic variables and should not be based on the status of surgical margins only.

## Figures and Tables

**Figure 1 animals-11-02367-f001:**
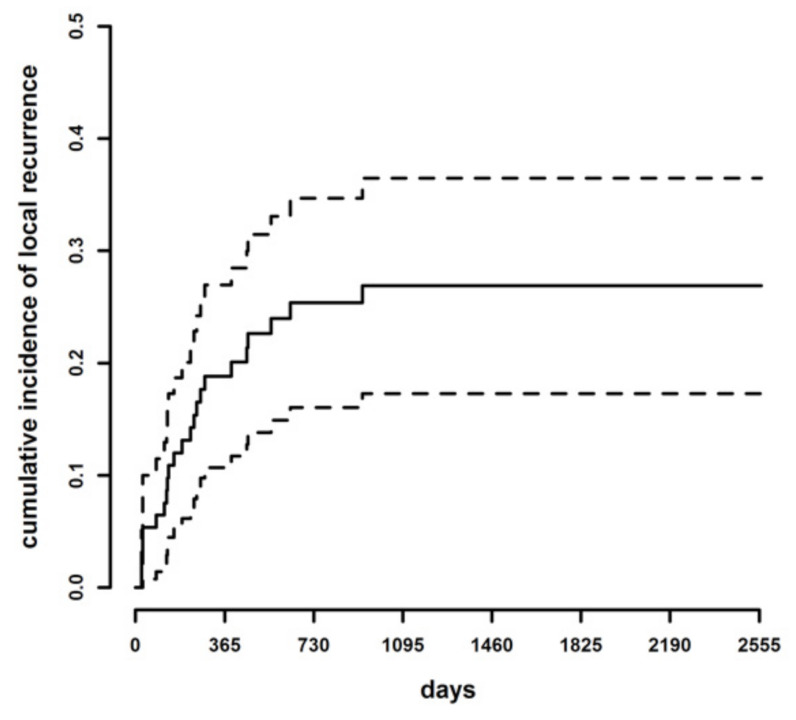
Cumulative incidence of LR (continuous line) and lower and upper limits of the 95% confidence intervals of the cumulative incidence curve (dotted lines) in the whole case series.

**Figure 2 animals-11-02367-f002:**
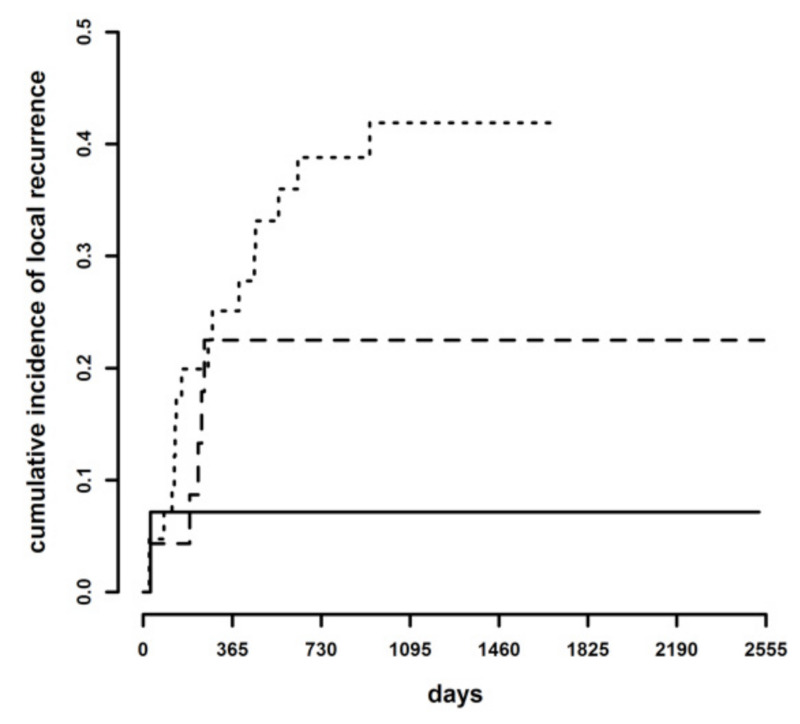
Cumulative incidence of local recurrence in tumor-free margins (continuous line), clean but close margins (dashed line), and infiltrated margins (dotted line).

**Figure 3 animals-11-02367-f003:**
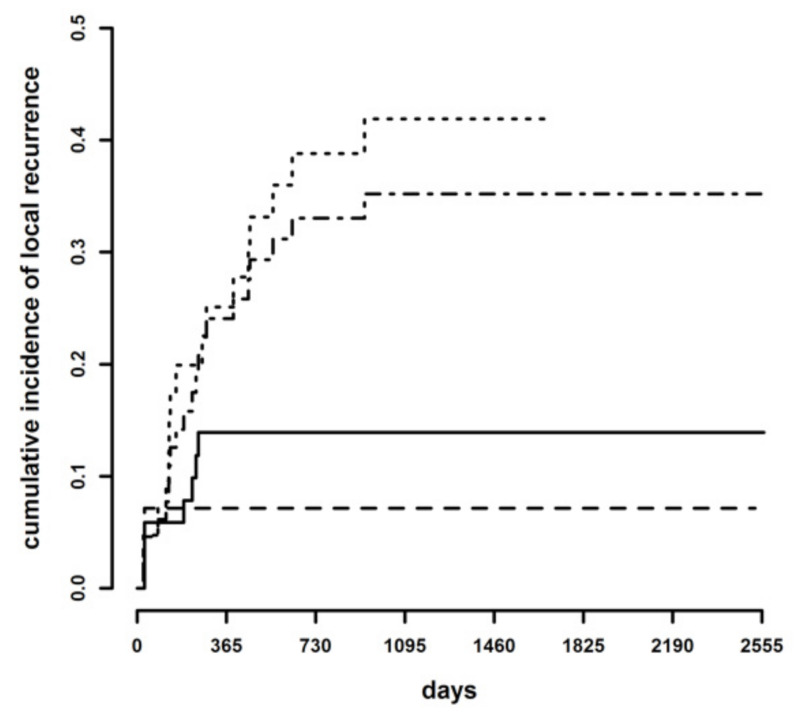
Cumulative incidence of local recurrence in tumor-free + clean but close (continuous line) margins versus infiltrated margins (dotted line), and in tumor-free (dashed line) margins versus clean but close + infiltrated margins (dotted-dashed line).

**Table 1 animals-11-02367-t001:** Signalment, clinical and histopathological characteristics of tumors, time to local relapse and outcome of dogs that developed local recurrence (LR).

Signalment	Location	Size	Histotype	Grade	Mitotic Index	Growth	Margins	Adj. Chemo	Time to LR (Days)	Outcome
		(Max cm)								(Days)
Mixed-breed, Fn, 13y, 17 kg	right PFL	10	PWT	2	11	expansile	Infiltrated	No	25	TUD (825)
Mixed-breed, F, 12y, 13 kg	right PHL	15	PWT	2	8	expansile	Infiltrated	No	283	TRD (1395)
Rottweiller, M, 7y, 40 kg	left DFL	NA	PWT	1	6	NA	Infiltrated	No	930	TRD (930)
Siberian Husky, Fn, 10y, 23, 5 kg	left PFL	10	PWT	1	4	infiltrative	Infiltrated	No	555	LFU
										−595
Mixed-breed, M, 12y, 14.5 kg	HN	4	PWT	2	12	infiltrative	Infiltrated	No	455	TRD (575)
Mixed-breed, Fn, 9y, 24 kg	right DFL	15	PWT	1	1		Clean but close	No	240	TUD (1123)
Mixed-breed, M, 14y, 10 kg	left PFL	3	PWT	3	10	expansile	Clean but close	No	30	LFU
										−30
Maltese, F, 8y, 3.5 kg	HN	7	PWT	2	6	expansile	Infiltrated	No	393	LFU
										−393
Labrador, F, 8y, 40 kg	left PFL	7	PWT	2	17	expansile	Infiltrated	No	634	TUD (2078)
Mixed-breed, Fn, 12y, 23 kg	left PFL	10	PWT	2	3	NA	Clean but close	No	250	TUD (300)
Pointer, Fn, 11y, 18 kg	left DFL	7	Rhabdomyosarcoma		3	NA	Infiltrated	Yes	266	LFU
										−266
Boxer, Fn, 11.5y, 32.4 kg	HN	3.5	Fibrosarcoma	1	2	NA	Tumor-free	No	30	TRD (60)
German shepherd, M, 11y, 40 kg	left DFL	10	PWT	2	12	NA	Infiltrated	Yes	117	TRD (190)
										(distant relapse 180 days)
Mixed-breed, M, 12y, 12 kg	Thorax	4	STS-NOS	2	3	infiltrative	Tumor-free	No	30	TRD
										−60
Dachshund, M, 3y, 12 kg	right PFL	7	PWT	3	40	infiltrative	Clean but close	Yes	225	TRD (229)
Whippet, Fn, 11.5y, 12 kg	right PFL	2	PNST	2	26	infiltrative	Infiltrated	Yes	85	TRD (200)
Italian Hund, F, 6y, 23.6 kg	NH	6	Mixosarcoma	1	1	infiltrative	Infiltrated	Yes	128	LFU
										−180
Mixed-breed, M, 8y, 44 kg	left PFL	4	PWT	2	5	infiltrative	Infiltrated	Yes	130	TRD (365)
Labrador r., Mn, 11y, 31 kg	left PFL	NA	PWT	2	6	expansile	Infiltrated	Yes	135	TUD (697)
Rottweiler, Mn, 10.5y, 37 kg	HN	3	PWT	2	1	expansile	Clean but close	No	190	TUD (402)
Mixed-breed, Fn, 12y, 36.1 kg	right PFL	4	PWT	1	1	infiltrative	Infiltrated	No	460	Alive (1061)
Boxer, Fn, 7.5y, 26.5 kg	left PFL	3.3	PWT	1	7	NA	Infiltrated	Yes	158	TRD (1144) (loco-reg relapse 1134 days)
Golden r., M, 10y, 32.4 kg	right PFL	5	PNST	1	2	infiltrative	Infiltrated	No	25	Alive (741)

Legend: F: female; Fn: neutered female; M: male; Mn: neutered male; PFL: proximal forelimb; DFL: distal forelimb; PHL: proximal hindlimb; DHL: distal hindlimb; TUD: tumor unrelated death; TRD: tumor related death; LFU: lost to follow-up; NA: not available.

**Table 2 animals-11-02367-t002:** Local recurrence sub-distribution Hazard Ratio (HR) for margins categories adjusted for each prognostic variable (results of Fine and Gray regression models).

	Infiltrated vs.			Clean but Close + Infiltrated		
	Tumor-Free + Clean but Close			vs. Tumor-Free		
Variables	HR	95% C.I.	*p*	HR	95% C.I.	*p*
**Margins**	4.36	1.79–10.7	0.0012	6.26	1.39–28.1	0.017
Location	6.85	1.78–26.4	0.0051	5.68	1.44–22.4	0.013
(other vs. extremities)						
**Margins**	2.77	1.12–6.85	0.028	4.21	0.932–19.00	0.062
Grading	2.64	1.12–6.20	0.026	2.5	1.066–5.86	0.035
(II + III vs. I)						
**Margins**	2.57	1.05–6.28	0.039	4.13	0.925–18.45	0.063
Adjuvant chemo	2.56	1.06–6.17	0.037	2.62	1.109–6.21	0.028
(yes vs. no)						
**Margins**	2.48	0.992–6.22	0.052	3.94	0.889–17.4	0.071
Size	1.08	0.956–1.21	0.22	1.08	0.968–1.2	0.17
(1 cm increase)						
**Margins**	5.046	1.431–17.79	0.012	7.52	0.913–62.01	0.061
Pattern of growth	0.823	0.288–2.35	0.72	1.19	0.443–3.21	0.73
(infiltrative vs. expansile)						
**Margins**	3.04	1.25–7.39	0.014	4.64	1.04–20.70	0.044
Mitotic count	1.05	1.00–1.09	0.038	1.04	1.01–1.07	0.024
(1 mitosis increase)						
**Margins**	3.065	1.252–7.50	0.014	5.411	1.273–23.00	0.022
Histotype	0.679	0.259–1.79	0.43	0.532	0.203–1.39	0.2
(PWT vs. others)						

Legend: C.I. Confidence Interval; *p*: Wald test *p*-value.

## Data Availability

The row data supporting the conclusions of this article will be made available by the corresponding author upon reasonable request, without undue reservation.

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
