# Peer review of "Surgical Margins in Canine Cutaneous Soft-Tissue Sarcomas: A Dichotomous Classification System Does Not Accurately Predict the Risk of Local Recurrence"

_animals, 2021, doi:10.3390/ani11082367_

Round 1

Reviewer 1 Report

Comments on the manuscript Animals- 1308315 entitled “Surgical margins in canine cutaneous soft-tissue sarcomas: a dichotomous classification system does not accurately predict the risk of local recurrence”, by Chiti et al.

This paper is a very interesting and relevant manuscript that highlights the need of standardization of margin assessment either macroscopically and/or histologically in STS. It shows the imperious requirement of guidelines regarding surgical margins evaluation of all types of animal tumors, not only STS, to achieve better survival rates, better follow up and/or therapeutic schedules, and to allow comparations between studies from different oncologic centers. In fact, nowadays, there is a lack of standardization in veterinary medicine and most studies use different classification systems, making comparison between studies difficult or even impossible, as authors enhanced in discussion section. So, studies on this field are welcome.

In general, the writing is clear, and results are very interesting. Nevertheless, there are some small points that need to be addressed to improve the comprehensibility of this manuscript.

Minor Compulsory Revisions

Simple Summary section

Line 15- Please amend ”… inlcuded…

Abstract section

Line 27- Please replace …”at first presentation were inlcuded…” by ”…at first presentation were included…

Lines 33-34- The p values presented here do not fit with p values present in Results section, lines 225-227, nor with those mentioned in figure 3 legend. These p values should be amended.

Introduction section

Line 72- Replace the sentence“…treated with surgery and adjuvant radiation therapy…” by “…treated with surgery and adjuvant radiation therapy (RT)…” in fact “RT” is later used many times in the manuscript.

Material and methods section

Line 86- Define FNA.

Results section

Line 164- Please clarify the following sentence: “…on the proximal limbs (50%), 24 (24.5%), 11 (11.2%) on the head and neck…” The body region of these 24 cases is missing…

Line 171- Define PWT on first mention.

Line 211- Please replace”with CbCM then tumor-free margins…” to ”with CbCM than tumor-free margins…

Lines 209-213: The p value between infiltrated and tumor-free margins (p=0.018) do not match with that present in figure 2 legend (lines 217-218: p=0.18). In fact, p values of “clean but close versus tumor-free margins” and “clean but close versus infiltrating margins” should be in the text not just in the figure 2 legend.

Lines 225-227: The p value between “tumor-free margins versus CbCM + infiltrated margins (p=0.33; HR = 5.05)” do not fit with that present in figure 3 legend, lines 230-231 (p=0.033). The p values of both “tumor-free margins versus CbCM + infiltrated margins” (p=0.33) and “tumor-free margins + CbCM versus infiltrated margins (p=0.11)” of Results section do not match with those p values present on Abstract section (p=0.01 and p=0.008, respectively).

Author Response

Reviewer 1

This paper is a very interesting and relevant manuscript that highlights the need of standardization of margin assessment either macroscopically and/or histologically in STS. It shows the imperious requirement of guidelines regarding surgical margins evaluation of all types of animal tumors, not only STS, to achieve better survival rates, better follow up and/or therapeutic schedules, and to allow comparations between studies from different oncologic centers. In fact, nowadays, there is a lack of standardization in veterinary medicine and most studies use different classification systems, making comparison between studies difficult or even impossible, as authors enhanced in discussion section. So, studies on this field are welcome.

In general, the writing is clear, and results are very interesting. Nevertheless, there are some small points that need to be addressed to improve the comprehensibility of this manuscript.

A: Thank you for the comments. The authors addressed all the points. 

Minor Compulsory Revisions

Simple Summary section
Line 15-
Please amend ”... inlcuded...

A: done

Abstract section
Line 27-
Please replace ...”at first presentation were inlcuded...” by ”...at first presentation were

included...

A: done

Lines 33-34- The p values presented here do not fit with p values present in Results section, lines 225-227, nor with those mentioned in figure 3 legend. These p values should be amended.

A: thank you for catching the error. All p-value were re-checked and now both abstract and text are correct.

Introduction section

Line 72- Replace the sentence“...treated with surgery and adjuvant radiation therapy...” by “... treated with surgery and adjuvant radiation therapy (RT)...” in fact “RT” is later used many times in the manuscript.

A: done

Material and methods section Line 86- Define FNA.

A: done – The abbreviation FNA has been removed considering the request of Reviewer 2 to reduce the number of abbreviations and since it was used just once.

Results section
Line 164-
Please clarify the following sentence: “...on the proximal limbs (50%), 24 (24.5%), 11 (11.2%) on the head and neck...” The body region of these 24 cases is missing...

A: done

Line 171- Define PWT on first mention.

A: done

Line 211- Please replace”...with CbCM then tumor-free margins...” to ”...with CbCM than tumor- free margins...

A: done

Lines 209-213: The p value between infiltrated and tumor-free margins (p=0.018) do not match with that present in figure 2 legend (lines 217-218: p=0.18). In fact, p values of “clean but close versus tumor-free margins” and “clean but close versus infiltrating margins” should be in the text not just in the figure 2 legend.

A: The correct p-value was 0.018 – to avoid repetition, all the p-value of the analysis has been added in the text and delete in the legend of the figure 2. 

Lines 225-227: The p value between “tumor-free margins versus CbCM + infiltrated margins (p=0.33; HR = 5.05)” do not fit with that present in figure 3 legend, lines 230-231 (p=0.033). The p values of both “tumor-free margins versus CbCM + infiltrated margins” (p=0.33) and “tumor-free margins + CbCM versus infiltrated margins (p=0.11)” of Results section do not match with those p values present on Abstract section (p=0.01 and p=0.008, respectively).

A: Thank you for catching the error. All the p-value have been re-checked.

Reviewer 2 Report

Reviewer’s comments and suggestions on manuscript: Surgical margins in canine cutaneous soft-tissue sarcomas.

General comments

This is an interesting paper aiming at evaluating a dichotomous classification system for the prediction of local recurrence of canine STS. The manuscript is well designed and have interesting findings that merit publication. The authors are using many abbreviated words that sometimes are hard to follow. The reviewer found 7 abbreviated words.  The manuscript needs a lot of English grammar and syntax editing.

My comments are listed below:

L.15, 25 at initial presentation were included

L.21, 36-37 The group CbCM may be considered as a separate prognostic category.

L. 56 STS report the use of qualitative criteria

L.64 as being inconsistently at risk…..

L. 67 risk of under- or overtreatment

L. 79 >0 mm

L. 85 at initial presentation

L. 90 Radiation Therapy (RT)

L.148 performed using a software

L.157 Spaniels

L.171-172 Perivascular Wall Tumors (PWT), and 20 (20.4%)were diagnosed as……….nerve sheath tumor

L.181 dogs, using one ….

L.189, 5 of which after detecting LR

L.190 before detecting LR

Table 1

Abbreviations at the end of the Table FN, MN- please clarify

Rhabdomyosarcoma

shepherd   

Figure 1 L.202 What these dotted lines represent please clarify

L.211 than tumor -free margins

L.224-227 You mentioned significance but the p=0.11 and the other p=0.33. These are not significand figures. Please amend

L.238 , while no significance was detected when the….

L.313 Please reword for clarity

L.364 rates between CbCM and ….

L.374-375 , histotype did not show significance for LR prognosis when evaluated ….

L.407 All authors have read ….

Author Response

Reviewer 2

General comments

This is an interesting paper aiming at evaluating a dichotomous classification system for the prediction of local recurrence of canine STS. The manuscript is well designed and have interesting findings that merit publication. The authors are using many abbreviated words that sometimes are hard to follow. The reviewer found 7 abbreviated words. The manuscript needs a lot of English grammar and syntax editing.

A: Thank you for your comment. Only one abbreviation (“FNA”) was removed because used just once. The authors hope that the revised version could be more readable and clearer.

My comments are listed below:

L.15, 25 at initial presentation were included

A: done

L.21, 36-37 The group CbCM may be considered as a separate prognostic category.

A: done

L. 56 STS report the use of qualitative criteria

A: done

L.64 as being inconsistently at risk.....

A: done

L. 67 risk of under- or overtreatment

A: done

L. 79 >0 mm

A: done

L. 85 at initial presentation

A: done – Recurrent STS has been added as an exclusion criterion to highlight the inclusion of the tumor at first presentation.

L. 90 Radiation Therapy (RT)

A: The abbreviation “RT” has been added in the introduction section as suggested by Reviewer 1.

L.148 performed using a software

A: done

L.157 Spaniels

A: done

L.171-172 Perivascular Wall Tumors (PWT), and 20 (20.4%)were diagnosed as..........nerve sheath tumor

A: done

L.181 dogs, using one ....

A: done

L.189, 5 of which after detecting LR

A: done

L.190 before detecting LR

A: done

Table 1

Abbreviations at the end of the Table FN, MN- please clarify

Rhabdomyosarcoma

shepherd

A: done

Figure 1 L.202 What these dotted lines represent please clarify

A: Dotted lines represents the upper and lower limits of the 95% coefficient interval. A specification has been added.

L.211 than tumor -free margins

A: done

L.224-227 You mentioned significance but the p=0.11 and the other p=0.33. These are not significand figures. Please amend

A: done

L.238 , while no significance was detected when the....

A: done

L.313 Please reword for clarity

A: Unfortunately, the number of the line does not correspond, and without any other reference, the authors have not been able to identify which statement should be reworded. If it is essential, the authors ask the Reviewer to kindly report the phrase so the authors could find it in the text.

L.364 rates between CbCM and ....

A: done

L.374-375 , histotype did not show significance for LR prognosis when evaluated ....

A: done

 L.407 All authors have read ....

A: done

Reviewer 3 Report

excellent work, clear and awesome

  1. This retrospective study investigates the impact of Clean but close margins (CbCM) on local recurrence (LR) of canine soft tissue (STS). Adjuvant treatments are recommended in dogs with incompletely excised cutaneous STS to reduce the risk of local recurrence LR, actually guidelines are lacking on how to manage clean but close margins (CbCM).
  2.  This is the first retrospective study aims at investigating the impact of CbCM on LR of surgically excised STS in a large mono-institutional cohort of dogs. This study hypothesized that CbCM would not display a significantly different behavior from tumor-free margins, and that a simplified and more objective dichotomous classification with a cut-off of histological tumor-free margins (HTFM) > 0 could better correlate with prognosis and thus be consistently applied for a simplified categorization of histological margins of canine STS.
  3.  Yes this paper is well written and easy to read.
  4.  The conclusion are very interesting for the future on STS.

Author Response

Reviewer 3.

excellent work, clear and awesome

This retrospective study investigates the impact of Clean but close margins (CbCM) on local recurrence (LR) of canine soft tissue (STS). Adjuvant treatments are recommended in dogs with incompletely excised cutaneous STS to reduce the risk of local recurrence LR, actually guidelines are lacking on how to manage clean but close margins (CbCM).

This is the first retrospective study aims at investigating the impact of CbCM on LR of surgically excised STS in a large mono-institutional cohort of dogs. This study hypothesized that CbCM would not display a significantly different behavior from tumor-free margins, and that a simplified and more objective dichotomous classification with a cut-off of histological tumor-free margins (HTFM) > 0 could better correlate with prognosis and thus be consistently applied for a simplified categorization of histological margins of canine STS.

Yes this paper is well written and easy to read.

The conclusion are very interesting for the future on STS.

A: The authors thank the Reviewer for the comments.